# Discharge Monitoring in Open-Channels: An Operational Rating Curve Management Tool

**DOI:** 10.3390/s23042035

**Published:** 2023-02-10

**Authors:** Michele Paoletti, Marco Pellegrini, Alberto Belli, Paola Pierleoni, Francesca Sini, Nicola Pezzotta, Lorenzo Palma

**Affiliations:** 1Department of Information Engineering (DII), Università Politecnica delle Marche, 60131 Ancona, Italy; 2Department of Life and Environmental Sciences (DiSVA), Università Politecnica delle Marche, 60131 Ancona, Italy; 3LIF Srl, Via di Porto 159, Scandicci, 50018 Firenze, Italy; 4Functional Centre, Civil Protection—Marche Region, 60126 Ancona, Italy; 5La Sibilla Società Cooperativa, Frazione Paggese 53, Acquasanta Terme, 63095 Ascoli Piceno, Italy

**Keywords:** environmental monitoring, rating curve, discharge data series, flood management, water level sensors

## Abstract

An aspect correlated with climate change is certainly represented by the alternation of severe floods and relevant drought periods. Moreover, there is evidence that changes in climate and land cover are inducing changes in stream channel cross-sections, altering local channel capacity. A direct consequence of a significant change in the local channel capacity is that the relationship between the amount of water flowing at a given point in a river or stream (usually at gauging stations) and the corresponding stage in that section, known as a stage–discharge relationship or rating curve, is changed. The key messages deriving from the present work are: (a) the more frequent and extreme the floods become, the more rapid the changes in the stream channel cross-section become, (b) from an operational point of view, the collection and processing of field measurements of the stage and corresponding discharge at a given section in order to quickly and frequently update the rating curve becomes a priority. It is, therefore, necessary to define a control system for acquiring hydrological data capable of keeping river levels and discharges under control to support flood early warnings and water management. The proposed stage–discharge management system is used by the Civil Protection Service of the Marche Region (east-central Italy) for the monitoring of river runoff in the regional watersheds. The Civil Protection Service staff performs stage–discharge field measurements using water level sensors and recorders (e.g., staff gauges, submersible pressure transducers, ultrasound and radar sensors) and a current meter, acoustic doppler velocimeter, acoustic doppler current profilers, portable mobile radar profiler and salt dilution method equipment, respectively. Power functions are fitted to the stage–discharge field data. Furthermore, extrapolation is performed to cover the full range of flow measurements; in general, extrapolation is not an easy task because of sharp changes in the stream cross-section geometry for very high or very low stages. In the present work, we also focused attention on the application problems that occur in practice and the need for frequent updating.

## 1. Introduction

Climate change has accelerated over the past five years as reported by the World Meteorological Organization (WMO), which, according to the European Centre for Medium-Range Forecasts Reanalysis version 5 (ERA5) [1] global climate reanalysis, declared the month of July 2019 as the hottest on record for the globe (WMO 2019). Signs of escalating climate change can no longer be ignored. The climate risk index (CRI) may serve as a red flag for existing vulnerabilities that may increase in seriousness as extreme events become more frequent or more severe due to climate change [2]. The CRI indicates a level of exposure and vulnerability to extreme events, which countries should interpret as warnings in order to be prepared for more frequent and/or more severe events in the future [2].

Flood risk and drought management are related to climate change; in this study, we focused on understanding the best existing and under-development monitoring, early warning, and water-use strategies. Climate change has led to concerns about increasing river floods resulting from the increased water-holding capacity of a warmer atmosphere [3]. These concerns are reinforced by evidence of increasing economic losses associated with flooding in many parts of the world, including Europe [3]. The occurrence of numerous large floods has exacerbated the concerns of increasing flood magnitudes. In Europe, a climatic-change signal in flood discharges over the past five decades was demonstrated in relation to changes in the timing of floods within a year [4]. The regional trends in river flood discharges in Europe (1960–2010) show increasing rainfall and soil moisture in northwestern Europe, decreasing rainfall and increasing evaporation in southern Europe, and decreasing and earlier snowmelt in eastern Europe. Most projections for the end of the twenty-first century suggest increasing floods in northwestern Europe due to increasing precipitation, and reduced floods in eastern and southern Europe due to increasing temperatures [3].

The stages of operational risk management can be divided into risk analysis, maintenance improvement, preparedness, and disaster response. The hazards must be combined with the vulnerability to the risk. The vulnerability of people or objects (the “elements at risk”) in a given area inundated by a flood of a certain magnitude is weighted by the frequency of occurrence of that flood. An accurate risk analysis process yields hazard or risk maps, which today are drawn by means of geographical information systems (GISs) based on extensive surveys of vulnerability combined with topographic maps. Residual risk remains a concern due to the failure of technical systems or to a rare flood exceeding the designed flood. When the risk maps are defined, strategic areas can be identified where monitoring is a priority. Here, control stations can be installed to acquire data to be used for analysis, modeling, and early warning [5]. In the framework of flood control, we focus on the aspect of river monitoring.

The second stage is the maintenance that must be planned, reported and improved after an accurate protocol is introduced (e.g., grass cutting, tree/bush work, channel maintenance, obstruction removal, defense repair and structure maintenance). The purpose of the third stage, called preparedness, is to reduce the residual risk through early warning systems and measures, which can be implemented to mitigate the effect of a flood disaster and support flood management [6]. In this step, historical discharge data are essential, but real-time recordings also allow us to evaluate the event and both forecast and nowcast the evolution through appropriate hydrological and hydraulic models. The final step of operational risk management is disaster relief, i.e., the set of actions to be taken when disaster has struck. This is the process of organizing humanitarian aid to the victims and later reconstruction of damaged buildings and lifelines [7].

The flood management system depends on three major factors: technology (hardware and software), financial resources, and urgency related to the risk map. These characteristics are location- and time-dependent. In this study, we focused on the technology used for data acquisition and data transmission, and on mathematical models used for water discharge estimation.

The empirical, or also theoretical, relationship existing between the water-surface stage (i.e., the water level), and the simultaneous flow discharge in an open channel (a type of flow in which one surface is free) is known as a stage–discharge relationship or rating curve (sometimes just “rating”). These expressions are synonymous, so they can be used interchangeably [8]. Rating curves are currently used in hydrology to estimate the discharge in natural and/or artificial open channels from water-level measurements. Thus, the traditional method to obtain current information on the discharge is to measure the water level with gauges and use the stage–discharge relationship to estimate the flow discharge, which is less expensive than direct and continuous discharge measurements. Many problems may afflict such measurements:The costs and human resources required for regular discharge measurements to develop and maintain the calibration of the rating curve;The rating curve is limited to the range of measured data;The rating is invalid if the cross-section changes;The discharge measurements typically scatter and do not show a unique relationship with the stage.

Loops and discontinuities in ratings may result from physical factors that affect any term of the equation describing the momentum of the flow that is not accounted for in the rating [9]. Stage–discharge rating curves for flow in rivers and channels are established by concurrent measurements of the stage *h* (direct measurement) and discharge *Q* (indirect measurement obtained from velocity acquisitions). The results are fitted to yield the rating curves [9]. When the rating curve is defined, it must be controlled to remain constant during the period. If the measured values substantially deviate from the rating curve currently used, they must be revised. For this reason, a validity range of the relationship consisting of a start date and an end date is defined. The end date will initially be unknown, but will be defined when a significant shift in the curve occurs and, thus, there is the need to define a new curve. Updates are especially needed after relevant flood events that could drastically change the river bed and banks.

Thus, an automated system is needed that primarily allows the definition of the rating curves. Then, in a graph, the time-trend of the following can be compared to understand whether the stage–discharge relationship is still valid or whether a rating shift occurs:Discharge measurements acquired in the field;Discharges estimated with the current rating scale;Discharges estimated with the new rating scale calculated, starting from the new measurements carried out in the field.

If a shift is present, then the exact point in time at which the curves merge must be evaluated: it represents the valid end date-time of the current rating scale and the valid start date-time of the newly calculated rating curve. We need to study methods and models to improve the data fitting between the discrete values derived from real measurements in the field (input) and the continuous data extrapolated from hydraulic models (output). A large portion of the modern practices used worldwide were developed by the United States Geological Survey (USGS) [10,11,12,13,14,15] and widely described over time by other scientific researchers [8], also using artificial neural networks [16,17,18], the World Meteorological Organization (WMO) [19,20], and ISO standards [21,22,23,24,25,26,27]. In Europe, hydrological data are collected by the Copernicus Emergency Management Service (CEMS) Hydrological Data Collection Centre (HDCC) and made available online at the European Flood Awareness System (EFAS) website. In 2011, EFAS became part of the CEMS initial operations in support of European Civil Protection. The operational components have been outsourced to Member State organizations. EFAS has been running and fully operational since autumn 2012. In October 2021, there were 68 data providers with 3949 registered stations in the dataset. HDCC data are representative of more than 50% of all the European water basins spread over 32 countries; approximately 20% of the stations deliver discharge data exclusively, another 20% only water level data and the rest provide discharge and water level data.

In Italy, since the second half of the 1800s, academics and the community of hydraulic engineers and practitioners have noted the need for a national service to be established to survey the characteristics of water courses; indeed, the Italian National Hydrographic and Mareographic Service (SIMN) was established in 1917. Over time, the hydrogeological risk map has been developed and updated by ISPRA (Institute for Environmental Protection and Research), as reported in Figure 1a. In the present work, we focused on the Marche Region territory (east-central Italy, Figure 1b), whose meteo-hydro-pluviometric monitoring network was managed until 2001 by the SIMN and then by the Functional Center of Regional Civil Protection Service. The whole monitoring network consists of two distinct networks: one mechanical (RM) and one telemetry (RT). The mechanical network that began working in 1916 was managed by the SIMN. The available sensors were thermometers, rain gauges, and hydrometers. The number, type (thermometric and/or pluviometric, and hydrometric), location, and period of operation of the stations have considerably changed over the years. The Civil Protection Service has been appointed since 2002 to perform the functions transferred from the SIMN. The activities related to data validation, processing, and publication are handled by the Marche Region Functional Center. The Marche Region is equipped with a telemetry monitoring system that was activated in June 2000. Starting from 2005, the discharges from some hydrometric sections have also been continuously estimated. The RT network, which definitively replaced the RM, in December 2022, mainly consists of 135 rain gauges, 121 thermometers, 107 hydrometers, 30 anemometers, 17 barometers, 113 hygrometers, 13 snow gauges, 18 sensors of incoming solar irradiation, and 6 soil moisture sensors. Flow measurement campaigns are also underway for estimating the discharge of the main regional rivers to define and update the rating curves in correspondence with important hydrometric sections. To date, the most-used software programs for collecting, storing, managing, validating, analyzing, and reporting water data are proprietary, file-based Hydstra and Water Information System (WISKI) produced by KISTERS [28]. To optimize data acquisition and update the rating curves based on field measurements, a nonproprietary web-based solution was developed for the Marche Region Civil Protection Service in the framework of the STREAM Project (strategic development of flood management).

From an operational point of view, quick and frequent updating of the rating curve at a given cross-section is becoming a priority; in this context, the software proposed in this work aims to provide an efficient stage–discharge management system.

## 2. Materials and Methods

The national and regional Civil Protection Services use different applications for monitoring flood risk. These applications have been modified over time with advances in software and hardware technologies for data measurement, management, and transmission. All the stations that record the hydrometric level are displayed in the operations center. Hydrometric rods, placed in the river, are used to verify that the electronic devices (gauge sensors) are always correctly calibrated. Having a redundant stage acquisition system is important; when both gauges provide a correct measurement, the rod value is typically taken as a reference due to its higher accuracy. The hydrometric rod (Figure 2) is positioned so that the zero value coincides with the gauge datum (G.D., presented in Figure 3).

The rating curves may be simple or complex depending on the river discharge, flow regime, and river bank and bed geometry. These relations are typically empirically developed from periodic measurements of stage–discharge using a hydrometric model with fitting algorithms. For high flow regimes, which are difficult to measure, a hydraulic model can be implemented for that river section. To verify the rating scale and monitor riverbed changes, periodic in situ flow measurements with acoustic Doppler current profilers (ADCPs) or velocimeters are required to control and keep the stage–discharge relationship updated. Moreover, hydrometers provide measurements of instantaneous values that may be affected by variations caused by local turbulence, waves, or obstacles under the sensors, so periodic surveys must be planned.

To obtain reliable measurements from hydrometric sensors, several measurements must be recorded in a short time interval, and the output value is assumed as the average between the instantaneous measurements. This operation is typically performed by an algorithm integrated into the sensor.

The data acquired by the control units are stored in a database, which is managed by the Marche Region Civil Protection Service and then transferred to the SIRMIP (Meteo-Hydro-Pluviometric Regional Information System) database where data are processed, validated, and then made available online [30]. Data recorded by the control units represent the “current data”, whereas only the periodic (i.e., averaged over a given time period) data are transferred to the SIRMIP, where two types of data are available: original (unvalidated) and validated data.

The fundamental assumption in stage–discharge analysis is that a unique discharge can be identified for any given stage [31]. The relationship between stage and discharge is defined by plotting discharge measurements (arranged on the abscissa axis) with the corresponding observation of the stage (arranged on the ordinate axis), considering whether the discharge is steady, increasing, or decreasing, and noting the range of change [24]. The plotting scale can be arithmetic or logarithmic. First, data validation is required to ensure that the recorded stages refer to the gauge data and that the calculated discharges are accurate. The number of direct flow measurements needed to develop a rating curve is defined by an ISO standard [24]: at least 15 or more measurements (for each defined segment) are needed, and they must be distributed over the entire range of the gauge height (also including the lower and higher extremes, which are useful in defining the correct shape).

As the number of stage–discharge measurements in the field increases, the accuracy of the hydrometric model used to determine the rating curve increases. The main problem is caused by the lack of stationary conditions on a river, which causes variations in the stage–discharge relationship.

In the database, three categories of rating curves are defined:Old rating curves: used in the past, but that are no longer valid due to riverbed changes over time;Current rating curves: currently used by the system to estimate the flow rate;New rating curve: created from the last new measurements determining the update of the curve if a substantial shift occurred.

To extract the data from the web application, the user first accesses the application with a username and password. Second, they select the variable of interest (for example, hydrometry). Once the variable has been chosen, the station search can be performed by selecting the basin, the municipality of interest, or via the interactive map. These options are useful when, with multiple extractions, the user needs to consider a series of stations over a certain area rather than a single station. Figure 4 shows the main interface created to manage the rating scales located in the database, the sensors of the regional network, and all the height–discharge measurements obtained in the field by civil protection volunteers.

The user may choose:All the historical series of the RT network;The hydrometric sensors of the RT network associated with at least one rating scale (official or unofficial);The historical series published in the Annual Hydrological Reports (annuals).

Once the sensor has been selected, the extractor offers two types of data:Original data: raw data without any processing;Validated data: automatically validated data, but still subject to changes during the processing of the annual report (manual validation).

Only the official data are published in the annual reports. The hydrometric data that can be extracted for analysis purposes are:Stage level: semihourly data at the stage level;Old discharge: semihourly discharge estimated from the corresponding stage level using the old rating scale;Current discharge: semihourly discharge estimated from the corresponding stage level using the current rating scale (also called the official rating scale);New discharge: semihourly discharge estimated from the corresponding stage level using the new rating scale;Stage-level min/max: minimum and maximum hydrometric levels recorded in the chosen interval. In case the official rating scale is available, it also provides the estimated discharge value;Stage level at 12: hydrometric level recorded at 12:00;Maximum discharge: if the official rating scale is available, it reports the maximum discharge recorded in the chosen interval;Average daily/monthly/annual discharge: if the official rating scale is available, it reports the average daily, monthly, and annual averaged flow rate;Rating scales: reports stage–discharge relationship estimating discharge from the stage level.

The velocity, depth, and width of flowing water change as the discharge increases at a particular river cross-section. Because river cross-sections tend to be roughly semi-elliptical, trapezoidal, or triangular, increasing the discharge results in increases in each of the other three factors. The typical relationships of width, depth, and velocity with discharge are described for various points on different rivers. These three factors generally increase with increasing discharge as simple power functions, and they are plotted as straight lines on logarithmic graphs [32]. Gauging stations are installed at river cross-sections where the stage–discharge relationship is as stable as possible to avoid constant changes in the rating scale. Specifically, a location is chosen, if possible, where discharges against gauge heights result in a smooth curve. The stage–discharge relationship for open-channel flow at a gauge station is governed by channel conditions downstream from the gauge, referred to as a “control” [24]. Thus, the control of a stream-gauging station refers to the nature of the channel downstream from the measuring section, which determines the stage–discharge relationship [32]. The physical characteristics of the channel, which govern the relationship between stage h and discharge Q at a cross-section, constitute the hydraulic control [33]. Two types of control can be defined: section control and channel control, depending on the channel and flow conditions. The main problem is that these controls may vary over time, reflecting a change in a segment of the rating curve. Section control is a specific cross-section of a stream channel, located downstream from a water-level gauge, that controls the relationship between gauge height and discharge at the gauge [8,24]. When the control is such that any change in the physical characteristics of the downstream channel has no effect on the flow at the gauging section, then such control is termed section control. In other words, any disturbance downstream of the control will not be able to pass the control in the upstream direction. Natural or artificial local narrowing of the cross-section, including waterfalls, rock bars, or gravel bars, creating a zone of acceleration, are some examples of section controls. A cross-section where flow does not accelerate or where the acceleration is insufficient to prevent the passage of disturbances from the downstream to the upstream direction location is called channel control. Channel control consists of a combination of physical features of the channel that dictates the river stage at a given section for a given flow rate. Such features include channel size, shape, curvature, slope, and roughness [34].

The controls can be categorized as natural or artificial controls. Natural section controls include rock ledges across a channel, the brink of a waterfall, or a local constriction in width (including bridge openings). All channel controls are “natural”.

Artificial section control is a control that has been specifically constructed to stabilize the relationship between the stage and discharge and for which a theoretical relationship is available based on physical modeling. These include weirs and flumes, discharging under free-flow conditions. In many cases, station controls are a combination of section control at low stages and channel control at high stages, and are thus called compound or complex controls [34]. At some stages, a combination of section and channel control is used for a short range. The segment of the rating curve where both controls are effective is commonly known as the “transition zone”, which is characterized by changes in the slope or shape of the stage–discharge relationship. In other situations, a combination of two different sections where each has a partial controlling effect may be possible. Careful field observations of the control state for each gauge are necessary to understand how time and site-specific factors might result in a rating curve change. Factors such as bedload movement, debris entrapment, bank erosion, seasonal aquatic and riparian vegetation changes, animal activities (e.g., humans and beavers), and ice effects must be considered when adjudicating gauging for rating curve development [31].

Depending on discharge, a river can work in three flow conditions:Low discharge: usually influenced by section control.Medium discharge: can be influenced by both types of controls. At some stages, a combination of section and channel control can occur.High discharge: usually influenced by channel control. The cross-section of a stream at a gauging station above a section control is not representative of the average discharge of the channel.

Therefore, in the final rating curve, using a single model is typically insufficient, so two to four must be defined, each of which defines a segment. The model used is the same, but the parameters that characterize it vary. The final rating curve is given by the union of the two to four segments that identify the different work areas. Indeed, due to the characteristics and dimensions of the rivers in the Marche Region, typically two to four segments are required.

All the acquired data, useful for river monitoring, have been collected in a database mainly composed of three tables: sensors table, measurements table and rating scales table.

The sensors table, on the database, was composed by:Sensor id: uniquely identifies the sensor located in the station;Station id: uniquely identifies the control station;Location name: identifies where the station is located;Gauge G.B. east: Gauss Boaga—Rome 40 (EPSG: 3004) east coordinate;Gauge G.B. north: Gauss Boaga—Rome 40 (EPSG: 3004) north coordinate;Gauge longitude (EPSG: 4326);Gauge latitude (EPSG: 4326);Gauge elevation: sensor position referred to the gage datum;Gage datum elevation: gage datum position referred to the mean sea level;Basin name;River name;Typical max value for low, medium, and high level: identifies typical stages range, useful for the operator to understand when take measurements according to the current height;Various notes.

The measures table was composed by:Measure id: uniquely identifies the measure;Station id: uniquely identifies the station;Measure date: the day of the measurement acquisition;Rod gauge height: hydrometric rod measurement;Sensor gauge height: ultrasound or radar sensor measurement;Discharge: discharge estimation on the field (*Q*);Instrument: type of appliance used to indirectly measure the discharge;Operator name;Measurements notes;Link of possible photos.

The rating scales table was composed by:Rating scale id: uniquely identifies each rating scale;Rating scale name: it is a code that identifies the segment of the rating scale. If the code was only 1, it meant that the rating scale was made up of a single segment, otherwise, it was made up of several segments identified by the same code;Rating scale validity onset (time and date): start of validity. It was determined by the hydrologist by analyzing the Q(t) curves;Rating scale validity end (time and date): end of validity. It was determined by the hydrologist by analyzing the Q(t) curves; when the rating scale was defined, we did not know until it would have been valid, and for this reason, a very distant default date has been set. Only when a new scale was defined was this value updated;Lower threshold: lower stage value where the current segment begins;Higher threshold: higher stage value where the current segment ends;Coefficient *C*;Exponent β;*e* (or *h*_0_): effective gauge height of zero flow;Producer: the name of who made the rating scale;Official data value (flag value): 0 (no) or 1 (yes);Updated gage datum (flag value): 0 (no) or 1 (yes);Updated relief (flag value): 0 (no) or 1 (yes);Updated hydraulic model (flag value): 0 (no) or 1 (yes);Note: for example, the type of reference stage measure (h-rod or h-sensor);Delivery date of the rating scale.

These three tables can be imported into the tool by the operators to be viewed and updated with the new data available.

## 3. Modeling Approach

The term uniform flow refers to the hydraulic condition in which the discharge, width, depth, cross-sectional area, and velocity are constant throughout the length of a channel. Perfectly uniform flow is rare in natural channels, but the condition is nearly true when the geometry of the channel cross-section is relatively constant throughout the course [35]. For an open channel, additional assumptions include:The depth of flow must be constant (that is, the hydraulic grade line must be parallel to the channel bed); this depth of flow is called normal depth.Because the velocity is constant, the velocity head does not change through the length of the section; therefore, the energy grade line is parallel to both the hydraulic grade line and the channel bed.

The physical structure of the channel control is linked to the shape of the rating curve through the hydraulic stage–discharge equation expressed by Equation (Equation 1) [13,14,15,21].
(1)$$Q=C*{\left(h-e\right)}^{\beta }$$

The parameters that define the relationship between the estimated discharge *Q* and the measured stage *h* (which represents the gauge height of the water surface referred to as the gauge datum) are:*e* (sometimes defined as $h_{0}$) is the effective gauge height of zero flow (or sometimes referred to as the cease-to-flow value [24]). This is an adjustment, sometimes called offset, that converts the stage level to the depth of water over the control.(h−e) is the effective depth of water on the control, sometimes called a hydraulic head.Coefficient *C* (sometimes defined in the literature as $Q_{1}$) is a scale factor numerically equal to the discharge when the effective depth of flow (h−e) is equal to 1, representing the product of the scale factor in the stage–area relationship and a flow resistance factor that includes the channel slope and the friction factor.Exponent β represents the sum of the shape exponent and the friction loss assumption exponent, being the slope of the rating curve when plotted on a logarithmic scale.

The conceptual model for the open channel has three prismatic geometries: a parabola, deep rectangle, and deep trapezoid. For each conceptualized shape, an offset must be estimated, where the offset is the elevation that contains all of the water within that specific shape. This conceptualization allows for the estimation of an exponent for each segment based on the geometric shape [31]. It must be estimated, to have a known parameter, using appropriate software, which, based on how the cross-section is created, provides the *e* parameters for the defined segments. When this extrapolation is not possible, then *e* is an unknown parameter that must be calculated (using fitting methods).

The rating curve calibration is an iterative process of the conceptual model using gauging measurements from the field. Knowing the shape of the cross-section, the offset *e* is defined, but it can vary with the stage increase. Thus, the number of segments needed to evaluate a reliable rating curve must be defined. For regular-shaped section controls, the effective gauge height of zero flow is nearly the same as the actual gauge height of zero flow, so it can be measured for the first segment of the rating scale by measuring the river depth at the deepest place in the control section as compared to the gauge datum, and then subtracting it from the gauge height *h* at the time of measurement [24]). At points where the control shape considerably changes, or where the control changes from section to channel control, the effective gauge height of zero flow usually changes. This results in the need to analyze rating curves in segments to properly define the correct hydraulic shape for each control condition. An example of the sensor used to acquire the water level on the field is reported in Figure 5, while all the heights are summarized in Figure 3a,b.

In these two schematized real applications, a sensor is positioned under a bridge and the river is modeled with a trapezoidal shape. Notably, the river heights are never measured in comparison with the stream bed of the river because it is not a stationary reference. Instead, the values report the distance of the water level from the gauge datum, which is an arbitrary altimetric benchmark whose zero level does not refer to any physical quantity. The distance between the gauge datum and the bed of the river level is not fixed, as the stream bed can evolve due to erosion and sedimentation. Conversely, the flow measurements are not affected by this problem.

To ensure the congruence of the measures over time, the measures must refer to a single immovable benchmark. However, the bottom of the river bed, by nature, undergoes continuous transformation. For example, during a flood, both the deposit of alluvial material with the relative raising of the riverbed and erosion with the consequent lowering of the bottom can be observed. For this reason, referring the measurements to a fixed quota is preferable to maintain the congruence and comparability of the values measured over time. The values displayed in the hydrometric level graphs do not indicate the real height of the water with respect to the bottom of the section considered, but the distance between the free surface of the water and the gauge. This measure is called gauge height, which is represented in Equation (Equation 1) as *h*. In general, a datum (also called gauge datum) is a point, plane, or surface by which systems of measurement are referred or related to one another. A vertical datum is a level surface to which elevations are referred, usually the mean sea level. $h_{G.D.}$ indicates the elevation between the mean sea level and the gauge datum chosen for the control station. $h_{m}$ represents the height between the sensor position and water level, $h_{s}$ is the height between the sensor position and the gauge datum (a known constant measured during the installation of the station), *e* is the previously described gauge height of zero flow, and (h−e) is the effective depth (or effective gauge height) used in the model of Equation (Equation 1).

If the elevation of the G.D. is lower than the stream bed elevation, *e* is a positive value, and the effective water depth is smaller than *h* (as reported in Figure 3a). Figure 3b shows the case in which the elevation of the G.D. is greater than that of the stream bed; *e* assumes a negative value, and then the effective water depth is greater than *h*.

When a gauging station is first established, the gauge datum should be set low enough to ensure that the lowest gauge height ever likely to be recorded while the stream is flowing at least 1 ft (30.48 cm) [36]. Negative gauge heights of zero flow are undesirable [36]. The gauge datum reference surface that is selected should be well below the estimated maximum scour depth to avoid negative gauge heights of zero flow over the life of the station.

The stream flow rate or discharge represents the physical quantity to be estimated using the rating curve. As previously stated, the first step consists of stage–discharge data acquisition in the field. Over the year, a number of measurements must be obtained to, for example, be able to homogeneously define the stage–discharge points in the range from the minimum to the maximum stage. Hydrologists, before defining rating curves, should analyze the accuracy of measurements, eliminating the discharges that are not well-fitted with the data from the analysis. Low and medium water measurements are normally obtained by using standard procedures [37,38], and their errors rarely exceed 5% [13]. Data preparation in the input is a fundamental aspect for the correct evaluation of the rating curve that is valid in the period under analysis. At least 12 to 15 measurements should be arranged during a given period of time [21]. These measurements should be equally distributed over the range of gauge heights, but this is not always achievable because the frequency of the various heights may differ (frequency of occurrence reduces as height increases) [8].

For regular-shaped section controls, the effective gauge height of zero flow is nearly the same. At points where the control shape markedly changes or where the control changes from section to channel control, the effective gauge height of zero flow usually changes. This results in the need to analyze rating curves in segments to properly define the correct hydraulic shape for each control condition. For most controls, however, the determination can be more exact by applying a trial-and-error method of plotting. An initial value is assumed, and measurements are plotted in logarithmic axes. If the resulting curve shape is concave upward (downward), then a somewhat larger (smaller) value for the effective gauge height of zero flow should be used. Usually, only a few trials are needed to find a gauge height of zero flow value that results in a straight line for the rating curve segment.

## 4. Data Acquisition

### 4.1. Sensors

Ultrasonic sensors (ULSs) can detect objects composed of different materials with millimeter precision, regardless of the shape and color. The sensor, managed by a microprocessor, bases its operation on an ultrasonic transducer, which sends a pulse to the surface whose distance is to be measured, then detects the resulting reflected echo. To minimize the systematic measurement errors usually occurring with ultrasonic sensors, the influence of variations in temperature (and, negligibly, in atmospheric pressure and relative humidity) on the speed of sound propagation in the medium must be considered. In the ULSs produced by ETG (Figure 6), this quantity does not influence the measurement of the hydrometric level because, by using an internal air temperature sensor, the variations in the speed of sound propagation are compensated.

The measurement principle of ultrasonic sensors is based on the analysis of the ultrasound travel time between sending and receiving or on checking the reception of the signal sent. The emitter and receiver are positioned in the same housing. The advantage is that even non-reflective or poorly reflective objects can be reliably detected. The sensors have been designed for continuous outdoor operation. The target below the sensor must be kept free from stones and various obstacles that could invalidate the measurement. The distance from the control unit must not exceed 200 m. The sensor must be horizontally positioned at a height with respect to the lowest level of the watercourse, not higher than its full scale; in addition, the highest level of the watercourse must reach no more than 50/100 cm depending on the sensor’s full scale. Finally, with respect to the vertical height of the sensor, an area is required that is free from obstacles, with a radius of at least 1.75 m. Once the sensor has been calibrated, to obtain a predetermined output with respect to a reference level, the sensor does not require special maintenance operations.

Additionally, other ultrasound (Figure 7) and radar sensors have been used on the stations to automatically acquire the stage. Their main characteristics are reported in Table 1.

Data from these sensors are compared with the acquisitions of the hydrometric rod in order to obtain a direct comparison to ensure the correct calibration. To speed up the process of reading the hydrometric rods, a camera is installed in each control station that frames the rod in order to provide the reading to the Marche Region Functional Center. This reading can be obtained by an operator or, even better, by means of suitable computer vision algorithms that are able to autonomously extrapolate the water level [39].

### 4.2. Application for Measurements and Calibrations

Another goal is to create an application for operators and civil protection volunteers to report any critical issues through photos captured with their smartphones in the field. Automatic water level detection systems provide various advantages, but may also provide incorrect values. Therefore, a transmission and warning system is needed to allow recalibration. Today, phone messages and emails are used.

Because many operators perform inspections with measurements, all measurements must be registered and managed by the developed system together with the volunteers. A database is also needed to archive the photos captured during the inspection, notes, stage value read on the hydrometric rod, etc.

Another important feature is knowing in advance the river section to visit for measurement inspection. Having a display that reports inspection priorities with different colors for each control unit would be useful. The preset rules must be define, for example, to assign priority when “a hydrometric level read in a station has never been measured in the last year” or “if no one has visited a control unit for more than 3 months”. These objective criteria are associated with a priority defined by a color scale to more quickly and efficiently organize teams and survey sites for flow measurement. The purpose of the application is to aid volunteers and technical operators to ensure everything converges in a single smart and portable application connected to the operation center.

### 4.3. Data Fields

Each control station is assigned a code, coordinates, name, and other data fields. It contains multiple types of sensors (one or multiple hydrometers, rain gauge, temperature sensor, etc.) with a unique code. All the tables in the database are linked to the sensor code. From the registry, all the codes of the hydrometers can be extracted, and then the actual data. In another table, the rating scales are stored, each linked to the sensor code.

For each hydrometer sensor, there are *n* different segments associated with the corresponding model parameters, depending on the river depth range (the structure is defined by Equation (Equation 1), but with different fitting parameters of *C*, β, *e*). Each segment is defined by the extremes of the lower and higher couple *h* and *Q*.

In the simplest cases, no segmentation has a unique rating curve, but this rarely occurs. Typically, up to a maximum of four segments are defined (when necessary to cover increased depths and irregular river banks) that together define the full rating scale. Thus, when updating the formulas, overlap must be avoided in terms of time range and stage segments. Having two or more rating scale segments with the same time range and the same stage segment on the database is not possible. Every 30 minutes, a value is sent to the SIRMIP database (a 40-character flag string is used), where all the information about the station for the year reported is kept. Three flags define the rating scale:–1: when the rating scale is not sent to SIRMIP;0: the scale is sent to SIRMIP, but not published in the annual report;1: the scale is sent to SIRMIP and is published in the annual report.

The data fields acquired and stored in the database are divided into three tables: sensors, measures, and rating scale tables (imported by the management tool in Figure 4).

### 4.4. Extrapolation and Interpolation

Interpolation between and extrapolation beyond gauging measurements can be misguided due to a lack of data. Because the measurements required to cover the upper and lower ends of the rating curve are often lacking, ratings are often extrapolated to estimate flows outside the range of observations [8]. Nevertheless, a large element of uncertainty exists in the extrapolation process; many methods of analysis exist to reduce the degree of uncertainty. The extrapolation of a curve beyond gaugings should not extend beyond the range explained by a conceptual model [31]. Interpolation can also be a problem for curves with sparse gaugings. In arithmetic coordinate graphs, the low point on the control is the gauge height of zero flow.

Even if flow measurements are recommended for values ranging from low to high flow, flood measurements are often missing. To overcome the problem and extrapolate the values used to define the upper part of the rating scale, the WMO has proposed four methods: conveyance slope method, areal comparison of peak runoff rates, flood routing, and step backwater method [20]. Researchers used the step backwater method exploiting the river modeling of HEC-RAS software [40]. This software allows the geometric shape reconstruction of the riverbed by analyzing different cross-sections. After integrating the measured data in the field with extrapolations from the hydraulic model developed in HEC-RAS, sufficient pairs of stage/discharge values can be generated to determine the rating curve.

### 4.5. Gauge Height of Zero Flow

One of the most important features in the stage–discharge relationship is the stage at the gauging station corresponding to a discharge close to zero, which is called the gauge height of zero flow. The “real gauge height of zero flow” is the gauge height of the lowest point in the control cross-section for section control. For natural channels, this value can sometimes be measured in the field by measuring the effective depth of flow at the deepest place in the control section, and subtracting the gauge height at the time of measurement (see Figure 3a,b). The effective gauge height of zero flow is instead a value that, when subtracted from the mean gauge height of the discharge measurements, causes the logarithmic rating curve to plot as a straight line [8]. Plotting the rating on an arithmetic scale, the effective gauge height of zero flow is the starting point of the gauge height in the first segment. Thus, when the rating curve consists of more than one segment, *e* can only be measured for the first segment, whereas the others must be estimated. Sometimes these data correction may be impractical, especially where the control is compound and channel control progressively shifts downstream at higher flows. The methods of assessing *e* used in the literature include [34]:Trial and error procedure;Arithmetic procedure (Johnson method);Computer-based optimization.

The first trial value of the data correction *e* is input either by the user based on a field survey or through the computerized Johnson method [19]. The Johnson method is based on expressing the data correction *e* in terms of the observed water levels. This is possible through the elimination of coefficients β and *C* from Equation (Equation 1) using a simple mathematical manipulation.

In computer-based optimization, the first estimate of *e* is varied to obtain a minimum mean square error in the fitting. This is a purely mathematical procedure and probably produces the best results on the basis of the observed stage–discharge data, but the result must be confirmed where possible by a physical explanation of the control at the gauging location. The procedure is repeated for each segment of the rating curve [34].

### 4.6. Uncertainty in Stage–Discharge Relationship

Uncertainty is a parameter associated with the result of the measurements that defines the dispersion of the samples that could reasonably be attributed to the measurement. Two types are possible:Standard uncertainty: defined by the standard deviation;Expanded uncertainty: defined by multiplying the standard uncertainty by a coverage factor indicated by *k*. If the distribution is assumed to be approximately normal (Gaussian), a level of confidence exists about 68% with k=1, about 95% with k=2, and about 99.8% with k=3.

The confidence limits must be evaluated for each segment of the estimated rating curve defined as the lower and upper limits, within which the true value is expected to lie with a specified probability, assuming a negligible uncorrected systematic error [41].

Uncertainty analysis requires a minimal number of 20 gaugings per segment, which, in practice, means the uncertainty computation is impossible for many extrapolated or not densely gauged segments [42].

The standard deviation of the residuals (or residual uncertainty) is expressed by Equation (Equation 2) [24].
(2)$$S=\sqrt{\frac{\sum_{i=1}^{n}{\left(ln\left(Q_{i}\right)-ln\left(Q_{c}\left(h_{i}\right)\right)\right)}^{2}}{n-p}}$$
where $h_{i}$ is the *i*th gauge stage measured in the field; $Q_{i}$ is the *i*th gauge discharge indirectly measured in the field; $Q_{c}$ is the gauge discharge estimated from the rating scale at the *i*th gauge stage; *p* is the number of rating curve parameters estimated from *n* gaugings. The standard uncertainty of the calculated value of $Ln(Q_{c}\left(h\right))$ at any stage *h* of the rating curve segment is expressed by Equation (Equation 3) [24].
(3)$$u\left[ln\left(Q_{c}\left(h\right)\right)\right]=S*\sqrt{\frac{1}{N}+\frac{{\left[ln\left(h-e\right)-\mu \right]}^{2}}{\sum_{i=1}^{n}{\left[ln\left(h-e\right)-\mu \right]}^{2}}}$$
where $\mu =\frac{1}{N}*\sum_{i=1}^{n}ln\left(h_{i}-e\right)$. The expanded uncertainty is defined by Equation (Equation 4) [24].
(4)$$U(Ln\left(Q_{c}\left(h\right)\right)=k*u(ln\left(Q_{c}\left(h\right)\right)$$
which has a typical value of *k* = 1.96 with a 95% uncertainty interval. Thus, we can define Equation (Equation 5) [24].
(5)$$C.I.=Q_{c}\left(h\right)*e^{\pm U(ln\left(Q_{c}\left(h\right)\right))}$$
where *C.I.* represents the confidence interval.

### 4.7. Riverbed Modelling Using HEC-RAS

In the field, a section of the riverbed was surveyed, inside which the section with the hydrometric rod is located. The river stretch to be studied varies according to the characteristics of the watercourse itself: the lower the slope of the riverbed in that section, the greater the length of the section to be studied. The distance between one section and the next was chosen on the basis of the variation in the characteristics of the riverbed or the presence of artifacts which could vary the flow rate. Usually, seven/eight sections in the river bed are measured. When, in the analysis, it was seen that the sections detected were insufficient to simulate the real trend of the discharge inside the riverbed, other sections have been added, also by interpolating those detected. The data of the river sections chosen to create a hydraulic model in HEC-RAS were surveyed in the field with GPS. These values were then inserted into the software so as to reconstruct the geometric shape of the riverbed and its surroundings.

For each section, it was necessary to define the riverbed and floodplain area. In the riverbed, the Manning’s roughness coefficient is usually smaller than that in the floodplain area. In fact, in the riverbed, the water flowing constantly keeps the bottom cleaner and, therefore, has low roughness. In the floodplain areas, on the other hand, the water flows only for flood regimes and, therefore, in an occasional way. The vegetation has plenty of time to regrow and make the roughness greater. Different roughness coefficients can be used, as well as for different areas of the section in the transversal direction, also for sections of the riverbed in the longitudinal direction. Within the geometric part of the modeling of the riverbed in HEC-RAS, we proceeded with the insertion of all those artifacts present in the section of the watercourse analyzed.

After entering all the necessary geometric characteristics, the project discharge values were defined (made up of the values measured in the section and the values of which we wanted to determine the corresponding hydrometric level). Here we entered those values that were difficult to measure in the watercourse or that were missing at the time of defining the rating scale. Finally, the boundary conditions have been defined which, in certain situations, can have a significant influence on the simulation result.

## 5. Algorithm and Block Diagram

The rating scale represents the linear regression curve that minimizes the standard deviations of the input points, acquired in the field and/or extrapolated with HEC-RAS software. The goodness of the result is evaluated using the $R^{2}$ metric. As previously defined, the global rating scale is derived from multiple segments, and each one is modeled using Equation (Equation 1). They must smoothly blend; as a whole, the rating scale must appear as a single curve with different slopes. The developed procedure to obtain the rating scale is schematized in Figure 8. The regression curve can be represented on an arithmetic or a logarithmic scale. On an arithmetic scale, the rating curve is concave downward, and extrapolating the necessary information is easier. On the logarithmic scale, the rating curve is represented by a straight line that intercepts the values of *C* and β as its slope. Typically, we have multiple straight lines with different slopes, equal to the number of segments used.

In the lower part of the rating curve, where the river-bed shape is more important, the values of β are usually greater than 2; in the upper part of the curve, where the geometry of the entire watercourse is more important, the values of β are usually less than 2. The constraint $\beta_{j}>\beta_{j+1}>1$ can be added, where *j* is the *j*th segment to have a final curve representation in which the solution of the mathematical regression on the fitting parameters is consistent with the physics of the model.

In the various segments obtained, discontinuities may exist between the end of one segment and the beginning of another (examples are shown in Figure 9 and Figure 10). When creating the complete rating curve, they must be eliminated to have unique correspondences between the stage and discharge.

The value of *e* is physically measured in the field for the first segment only, whereas for the other segments, the value is analytically derived through iterations, as shown in Figure 11. In the latter case, the best value of *e* is given by the global maximum, which means the maximum value of the $R^{2}$ in the segment under consideration.

Starting from Equation (Equation 1) with the data measurements from the field, the value of *e* in the first segment is measured, whereas the others *e* are calculated using an iteration loop, where the value of *e* is changed from a minimum value (value in the first segment) to a maximum value. Thus, using the coefficient of determination ($R^{2}$) as an evaluation metric, the best values of *e*, *C*, and β can be found. The offset value can be positive, as depicted in Figure 3a, or negative, as presented in Figure 3b, depending on how the gauge data are positioned. Over time, changes in the cross-section can occur and cause the value of *e* to change from negative to positive or vice versa. In these cases, the value must be updated through measurement in the field. Inputting the value of *e* in Equation (Equation 1), the best segmentation that fits the input data can be derived. These parameters obtained from the mathematical regression must be physically plausible in relation to the analyzed cross-section.

The discharge *Q* must always be positive, so the constraint is (h−e)≥0:h=e, Q=0 is the first point in the rating curve;h>e are the other points.

Higher stage values $h_{max}$ on the rating scale can be represented by the maximum height value recorded in the annuals. As previously described, if discharges near the maximum height cannot be measured, for either practical reasons or lack of events, they can be extrapolated with HEC-RAS software.

The standard procedure used to determine the full rating scale is shown below. The idea is to compose the rating scale by creating segments of curves, each with its own parameters defined by Equation (Equation 1), in order to have the smallest difference compared with the measured input points. The complete rating curve therefore consists of one to a maximum of four segments. The accuracy depends on the number of measures available in a defined range of time, measurement errors, and how they are spread all over the range.

These are the steps used to evaluate the rating scale:As a starting point for the linear regression, at least 15 input data are required from field measurements and/or simulated values from HEC-RAS software, all obtained during the period of analysis [21]. The calculations are based on the Bernoulli equation, which is performed step-by-step from downstream of the modeled reach to the cross-section of the gauging station. The measurements are sorted in ascending order according to their date and time; such measurements should be equally distributed over the range of gauge heights, but this is not always achievable in the field because the frequency of the various heights may differ (frequency of occurrence reduces as height increases) [8].Having at least 15 measured points is preferable to take the *m* oldest points in time and sort them in ascending order with respect to the stage.The entire group of points of the input data is divided into two segments using iterations to find the best subdivision. It is possible to define *n* = minimum number of starting points, and *m* = number of total points. By point, we mean the stage–discharge pair. The iterations start from the first segment composed of the first 3 points (n=3), which define the behavior near the bottom of the riverbed, and the second segment starts from (n−k) to *m* elements; *k* identifies the overlap of points between the two segments, which is used to reduce the discontinuities previously defined when segments are being connected. *k* can be set to 3 values:
k=−1, during the iterations, no superimposition occurs, and the elements in the second segment run from (n+1) to *m*;k=0, during the iterations, only one point is superimposed, and the elements in the second segment run from *n* to *m*;k=1, during the iterations, two points are superimposed, and the elements in the second segment start from (n−1) to *m* (this is the default configuration chosen because it smoothly anchors the two segments together).*n* is increased by one for each iteration. The subscript *j* indicates the segment created whose points vary during the iterations indicated by the subscript *i*. At the end of the loop (with a total i=m−n iterations), it is possible to define, for segment *j* and segment j+1, the best segmentation that maximizes the coefficient of determination $R_{mean}^{2}$. R-squared is the statistical metric that measures the proportion of the variance in the dependent variable that is predictable from the independent variable in the regression model.The best values of the fitting parameters for these segments are calculated using linear regression (the modeling technique used to create the stage–discharge relationship). To apply linear regression, Equation (Equation 1) must be represented in logarithmic terms, as shown in Equation (Equation 6).
(6)$$ln\left(Q\right)=ln[C*{\left(h-e\right)}^{\beta }]$$
Applying the properties of logarithms, the relationship in Equation (Equation 7) is obtained.
(7)ln(Q)=β∗ln(h−e)+ln(C)
which is the same structure as that of the equation of a straight line, as defined by Equation (Equation 8).
(8)y=m∗x+qBoth *x* and *y* are affected by an error, but the error in *x* can be assumed to be negligible. Then, a fitted linear regression model of the response *y* (that is, ln(Q)) to the input data *x* (that is, ln(h−e)) is obtained, and the fitting parameters β=m and $C=e^{q}$ are derived.A stage array is created starting from the lower stage value (equal to *e*) to the maximum (equal to $h_{max}$) with a resolution between each stage sample that we chose as equal to 0.001 m. Then, inputting this array and the estimated coefficients in Equation (Equation 1), we find the discharge fitted array. The root mean square error (RMSE) measures the model’s accuracy and summarizes how closely the estimates of the model match the observed responses (residuals) in the considered segment. The measure is always positive or zero, with zero being the best result possible. To calculate the RMSE, the observed responses ($Q_{i}$) and the predicted responses (${\hat{Q}}_{i}$) are used from the model on the corresponding inputs ($h_{i}$). The RMSE is expressed by Equation (Equation 9).
(9)$$\mathrm{RMSE}=\sqrt{\frac{1}{n}*\sum_{i=1}^{m}{\left(Q_{i}-{\hat{Q}}_{i}\right)}^{2}}$$After the segmentation and parameters definition, the exact beginning and end of each segment must be found. This is achieved using a function that identifies the beginning and end of each segment, evaluating the minimum difference between the curve’s superimposition. This is performed to reduce the gap between the segments and to create as smooth a final curve as possible without discontinuities. The fusion points, also called breakdown points, are used to create the final smoothed rating curve.The complete final rating scale, if considered valid, is plotted and used as a reference to extract the discharge estimation. When joined, the separate curves form a smooth continuous combined curve. Usually, the discharge (dependent variable) is plotted as the abscissa so that the concavity is downward (in the arithmetic plot).Once the rating scale is constructed, all subsequent measurements in the field are compared with the rating scale in force. In the event of substantial changes, the curve must be updated.The stage–discharge relationship can shift because of variations in the physical features of the gauging station control. Thus, we need to ensure that the rating scale chosen as a reference is stable over time. The time period over which this shift occurs is referred to as a period of shifting control. During these periods, frequent discharge measurements are needed to identify the new stage–discharge relationship. The main issue in defining the rating curves is precisely given by the constant shift check of the measurements in the field with respect to the curve in force. As stated by the ISO normative [24], if new acquisitions in the field show minimal variations, then the rating scale remains valid; if the shifts begin to be notable, the scale must be updated with the new measurements. The percentage by which a measurement may deviate from the rating curve without applying a correction is usually based on the uncertainty in the discharge measurement. If, for instance, most discharge measurements have up to 5% uncertainty, then shifting-control techniques do not need to be employed unless a check measurement is further than 5% from the rating curve. Another approach to check the shift is performing a statistical analysis of the rating curve to define the dispersion (standard deviation) of the measurements around the rating curve. When two or more measurements indicate a deviation of more than two standard deviations from the rating curve, then a shifted curve or a new rating curve needs to be defined [24]. The standard deviation is usually separately defined for each segment of the rating curve. Shift curves usually have a shape that is similar to that of the original rating curve [24].

This represents the developed procedure for the determination of the rating curve.

## 6. Results

Cross-sections relating to three different rivers were analyzed and the relative rating scales were obtained using the proposed tool. The Chienti River is one of the longest waterways in the Marche Region, which is located in the south of Ancona. It originates from Serravalle di Chienti (MC) and flows into the Adriatic Sea, between Civitanova Marche and Porto Sant’Elpidio, covering approximately 91 km. The investigated area was approximately about 525 meters along the course of the Chienti River in the village of Pontelatrave, municipality of Valfornace (MC), where the hydrometric station was located (Figure 12a). This underlies a catchment area of approximately 233.97 m^2^.

We found seven river sections useful for modeling the watercourse (Figure 12b).

A hydrometric rod was located in Section 2, where the level of the water surface was read (Table 2).

The riverbed was located at 0.005 m compared with the gage datum. Thus, e=0.005 was the value we used in the first segment. With the measured points and those determined by the model in HEC-RAS (where the river sections measured with GPS were used), the scale obtained with the previously explained procedure was composed of three segments, defined in Table 3.

The result is represented in Figure 13. If we magnify the area between each segment, some discontinuity, which was smoothed and plotted in the final step, can be observed (Figure 14).

We considered another case study for the Nera River (Figure 15), which is located in the mountainous part of the Marche Region. It is the only river with a source in the Marche, but flows into the Tyrrhenian Sea after having joined the Tiber River. The Nera River originates in Castelsantangelo sul Nera (MC); after passing through the municipality of Visso, it enters the nearby region of Umbria. The measurements recorded over the years are shown in Table 4, which compares the discharge of the rating scale currently used by the Civil Protection Service and the discharge evaluated using the proposed rating curve.

The bottom of the riverbed is located at −0.35 m compared to the gage datum. Thus, e=−0.35 was used in the first segment. With the measured points and those determined by the model in HEC-RAS (where the river sections measured with GPS were used), the scale obtained with the previously explained procedure is composed of three segments defined by Table 5. The results are represented in Figure 16 and Figure 17.

The last rating scale evaluated was for the Esino River, which is located in the central part of the Marche Region. The Esino River originates from the nearby Umbria Region, and flows into the Adriatic Sea. The measurements recorded over the years, in the Camponocecchio station (Figure 18), are shown in Table 6, which compares the discharge of the rating scale currently used by the Civil Protection Service and the discharge evaluated using the proposed rating curve. With the measured points and those determined by the model in HEC-RAS, the scale obtained with the previously explained procedure was composed of two segments, defined in Table 7. The results are represented in Figure 19.

## 7. Discussion

The automated rating curve evaluation that we developed in this study allows us to obtain the most accurate possible rating curve with the available input data. The proposed procedure enables the determination of a rating scale by dividing it into different segments. The estimated segments are smoothly and coherently connected to each other. The results showed an optimization of the estimate, which results in a smaller error than that of the rating curves currently used by the Civil Protection Service. The management tool (Figure 4) enables the administration of all the control stations in the Marche Region, allowing the viewing of the history, creating new rating curves, saving new data of measurements from the field, and updating the databases.

This locally developed approach to river monitoring can also be extended to other regions using this open-source system that allows even low-level management.

## 8. Conclusions

Having a rating scale is essential to estimating the flow rate through a section of a watercourse at any time, knowing the water level. The validity of the discharge–water level relationship must be frequently checked and, if necessary, updated as soon as possible to obtain coherent flow data series.

Owing to the knowledge of the estimated discharge, the Functional Center of Civil Protection has the ability to calibrate the numerical model with the historical data series, monitoring the river flow through water-level sensors in telemetry, promptly issuing the necessary warnings in the forecasting phase and supporting emergency managers during a flood event.

Data preprocessing is a fundamental step because, if the input data measured in the field and extrapolated from HEC-RAS are updated over time and have a negligible error, the model will produce a more accurate estimate. The flow rate values are also fundamental for both determining the hydrological balance of a basin and estimating return periods from the historical series, essential for designing hydraulic works. The complexity of determining an adequate scale is caused by the need for the knowledge of hydraulics, river dynamics, banks and riverbed geometry, statistics, and geomatics. The discharge data indirectly measured from the riverbed to extrapolate the rating scale are sometimes lacking or technically difficult to obtain. With appropriate settings and surveys of river sections that are more or less updated, the watercourse can be reconstructed, sometimes quite faithfully. With these new data, together with the available measurements, the rating scale can be determined, extrapolating the highest values that are more difficult to measure. The proposed procedure allows us to automatically derive the model used to estimate the discharge, contributing to a faster and easier updating of the rating curve with the management tool.

## Figures and Tables

**Figure 1 sensors-23-02035-f001:**
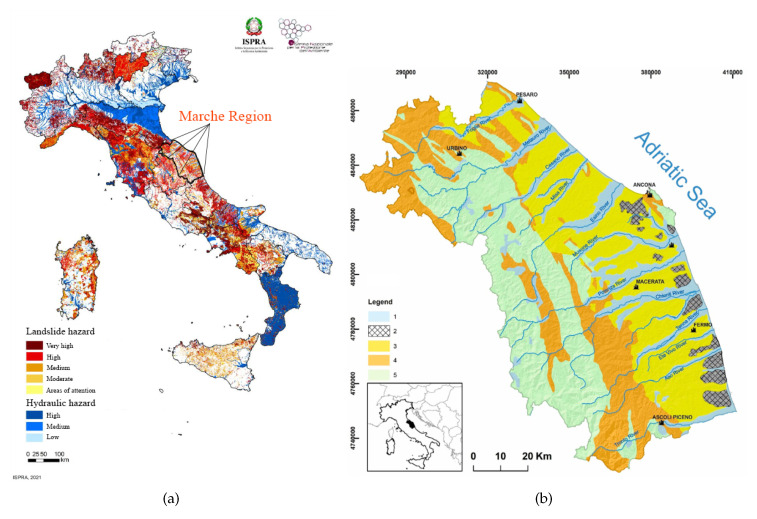
Area subjected to control. (**a**) Map of hydrogeological risk in Italy, referred to the ISPRA report 2021. (**b**) Schematic geological map of the Marche region. (1) Main continental deposits (Pliocene-Pleistocene-Holocene); (2) sands and conglomerates (Pliocene–Pleistocene); (3) clays and sands (Pliocene–Pleistocene); (4) arenaceous-marly clayey turbidites (late Miocene); (5) limestones, marly limestones and marls (early Jurassic–Oligocene) [29].

**Figure 2 sensors-23-02035-f002:**
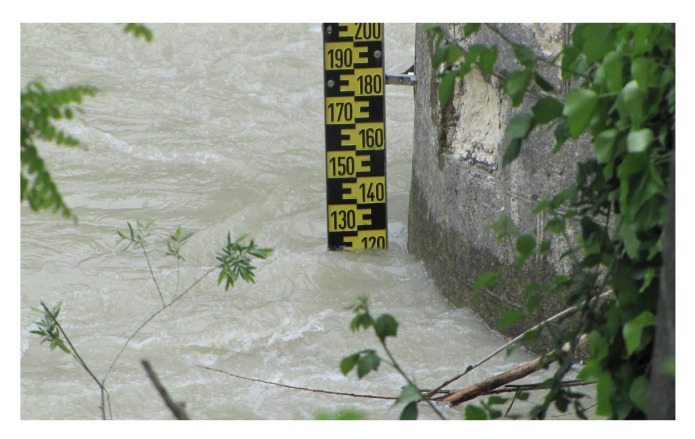
Example of hydrometric rod installed by the Civil Protection in the municipality of Acqualagna (PU), Candigliano basin. The hydrometric rod (also called staff gauge [5]) is made of anodized aluminum with anti-corrosion treatment to resist even environments with contaminants and abrasions. The high color iridescent effects allow excellent visibility of scale even at a considerable distance and in any application (concrete, bricks, soil, etc.).

**Figure 3 sensors-23-02035-f003:**
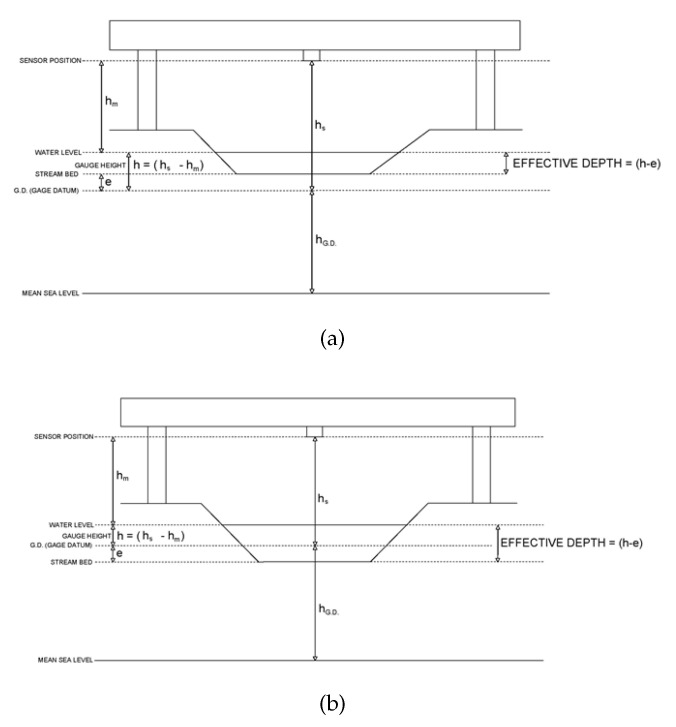
Gauge sensor applied under bridge to measure river stage and depth. (**a**) In this case, G.D. < stream bed. (**b**) In this case, G.D. > stream bed.

**Figure 4 sensors-23-02035-f004:**
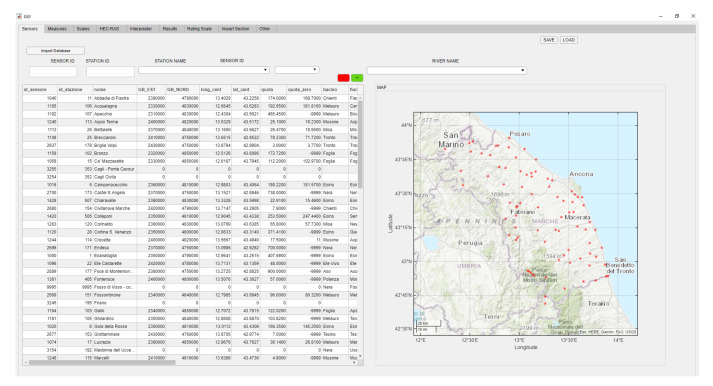
Management tool for acquiring, processing, and saving rating scales. Red dots indicate sensors positioned in the field for automatic acquisition of the water level.

**Figure 5 sensors-23-02035-f005:**
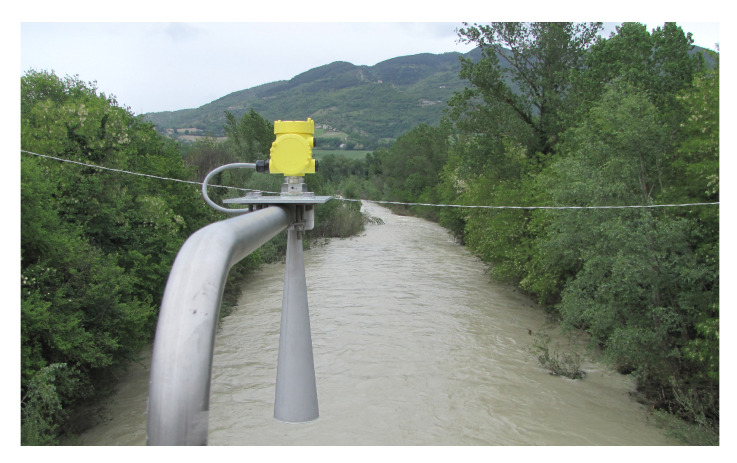
Example of gauge radar sensor used to acquire stage measurements (VEGAPULS SR 68, Table 1).

**Figure 6 sensors-23-02035-f006:**
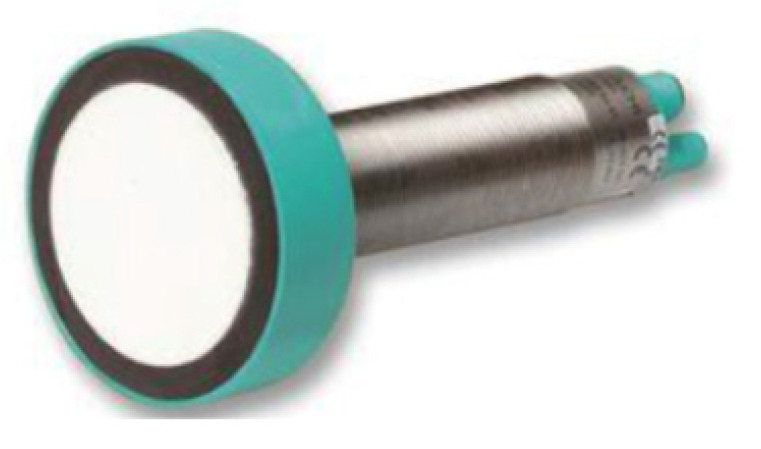
ETG ultrasound sensor (Table 1).

**Figure 7 sensors-23-02035-f007:**
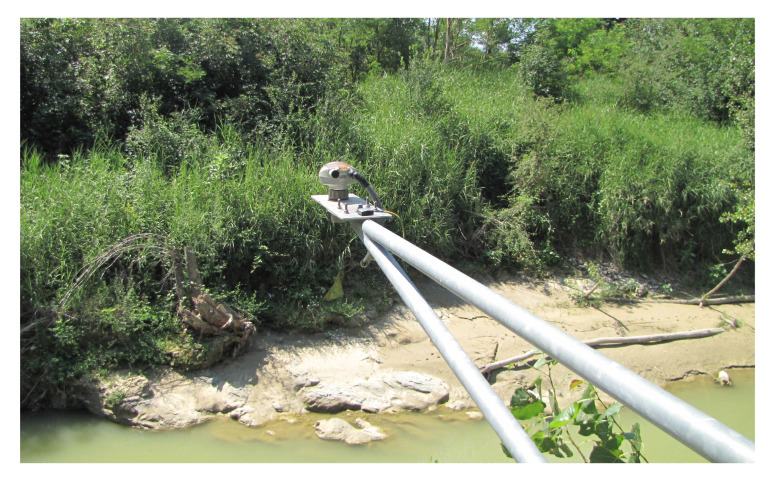
Example of a gauge ultrasonic sensor used to acquire the stages measurements (SITRANS Probe LU, Table 1).

**Figure 8 sensors-23-02035-f008:**
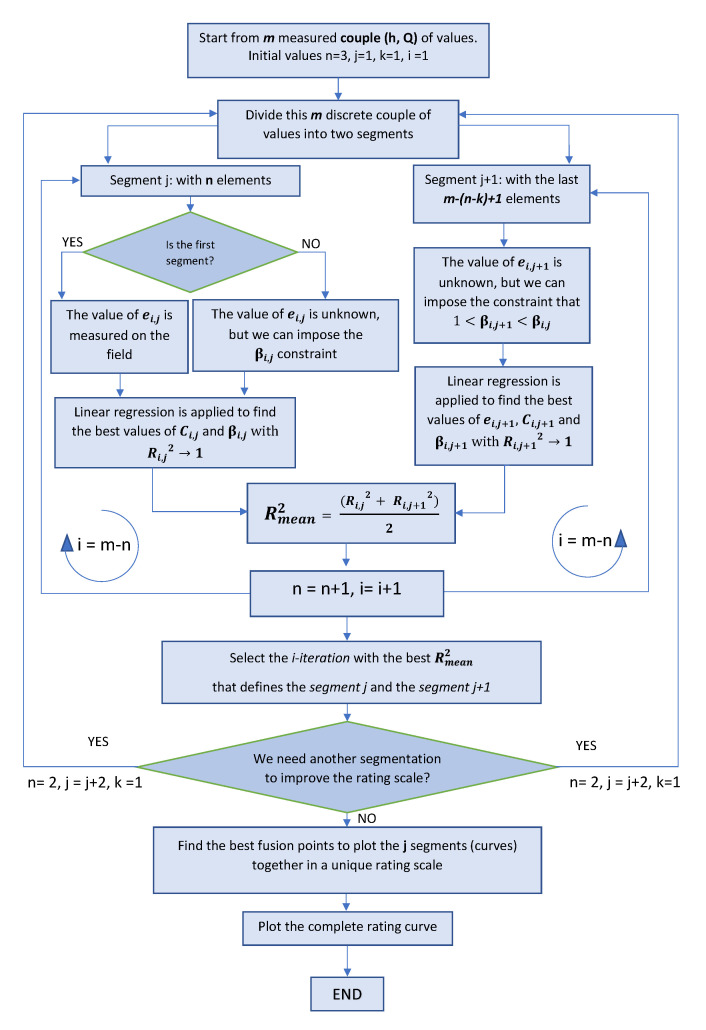
Block diagram: procedure developed to create rating scales.

**Figure 9 sensors-23-02035-f009:**
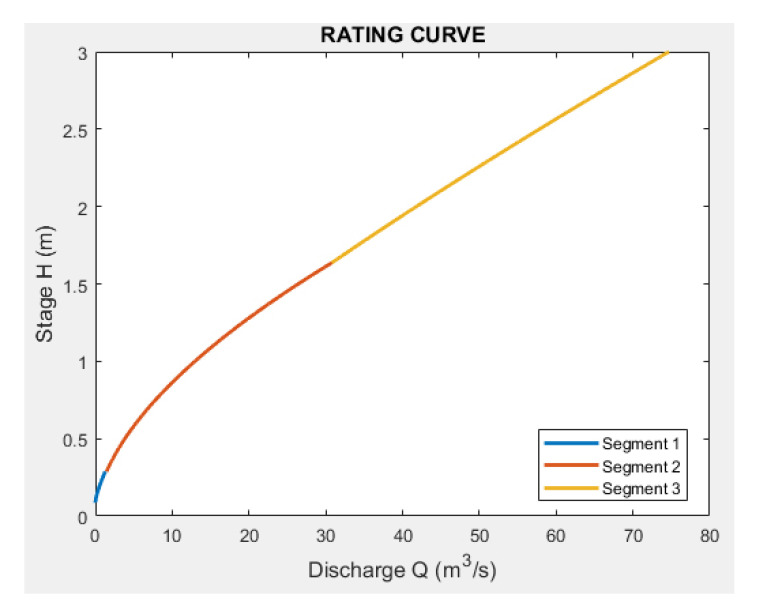
Example of rating curve composed of 3 segments.

**Figure 10 sensors-23-02035-f010:**
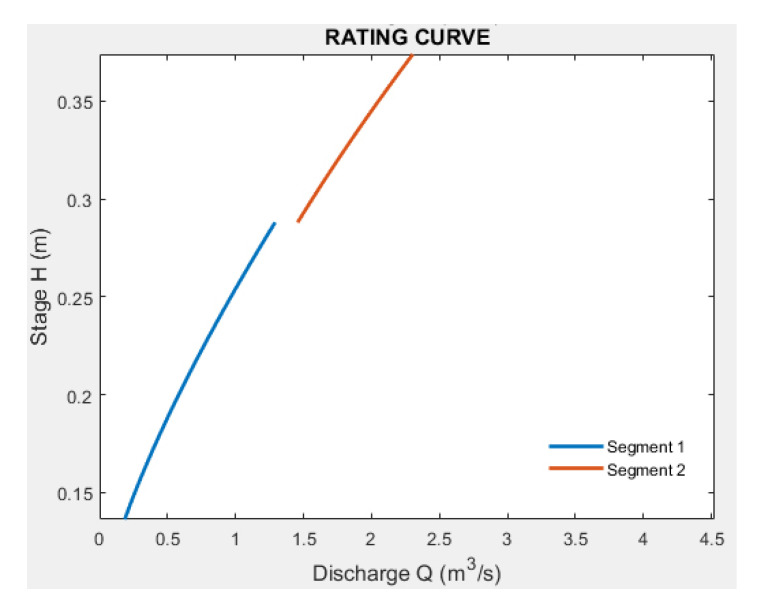
Magnification of previous figure showing the discontinuity that produces two different discharges at the same stage value.

**Figure 11 sensors-23-02035-f011:**
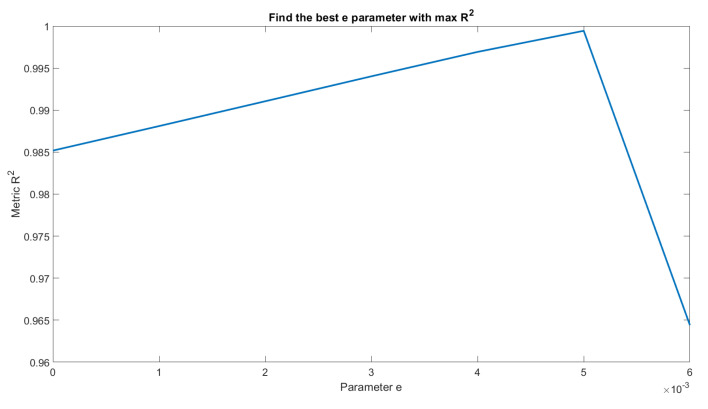
Example of a graphical representation of the iteration result. Value of *e* is varied from a minimum to a maximum value and is taken as the value that maximizes $R^{2}$ in the segment under consideration.

**Figure 12 sensors-23-02035-f012:**
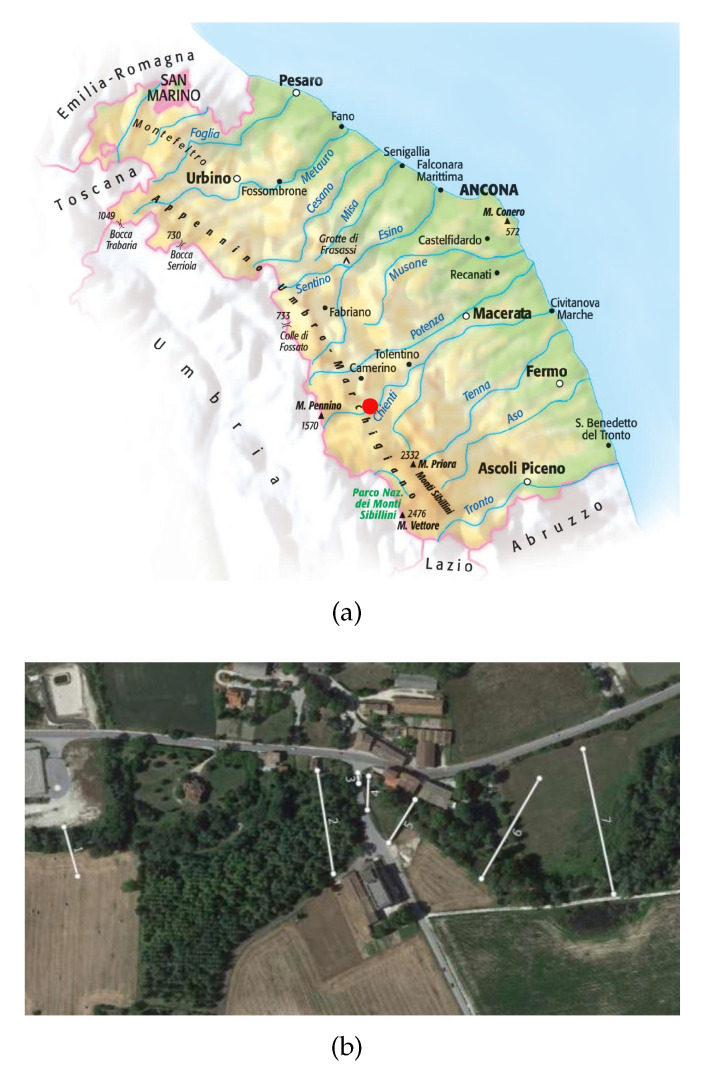
Information about the station located along the Chienti River in the village of Pontelatrave. (**a**) Marche Region, where the red dot indicates the investigated area of the Chienti River near the village of Pontelatrave, in the municipality of Pievebovigliana. (**b**) Schematization of the Chienti River and sections surveyed in the study area.

**Figure 13 sensors-23-02035-f013:**
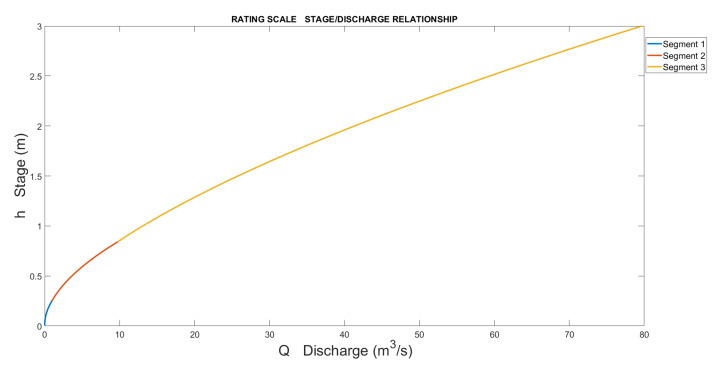
Rating scale of Pontelatrave with the 3 segments.

**Figure 14 sensors-23-02035-f014:**
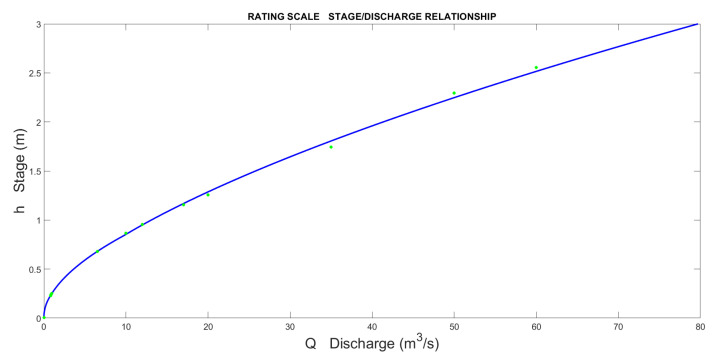
Complete rating scale of Pontelatrave with all the input points used for regression.

**Figure 15 sensors-23-02035-f015:**
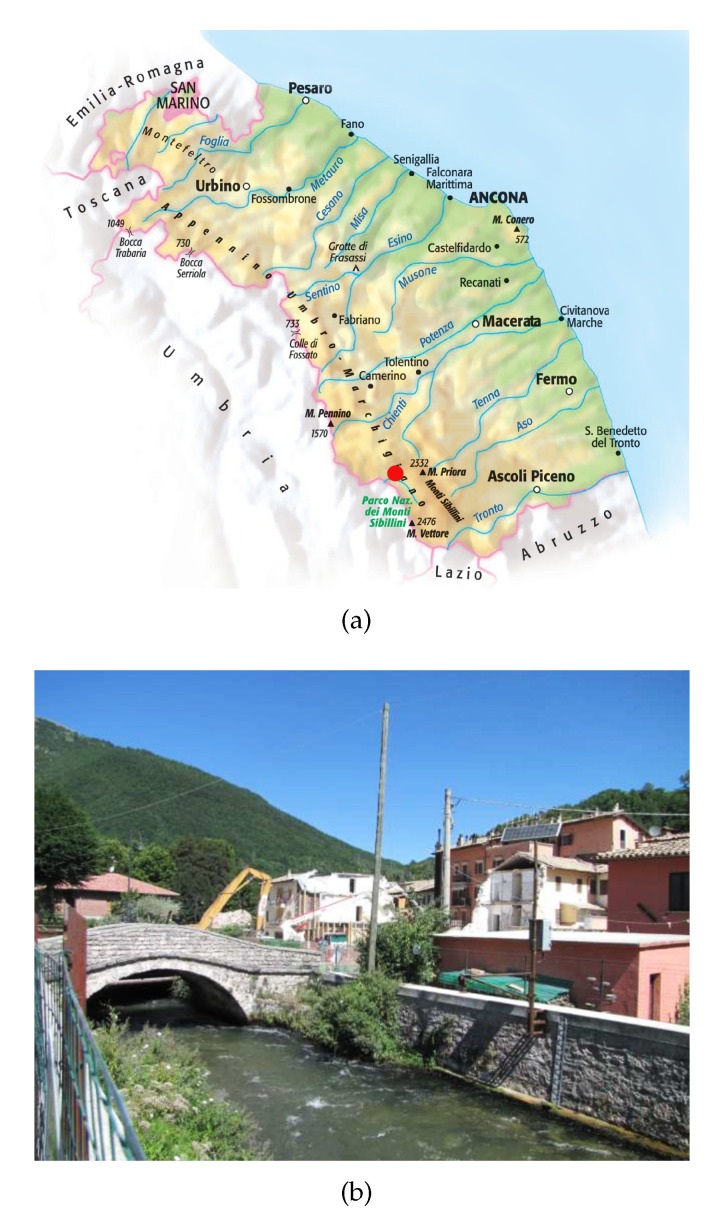
Information about the station located along the Nera River, municipality of Visso (MC). (**a**) Marche Region, where the red dot indicates the investigated area of the Nera River, in the municipality of Visso. (**b**) Hydrometric rod position on the Nera River located in Visso.

**Figure 16 sensors-23-02035-f016:**
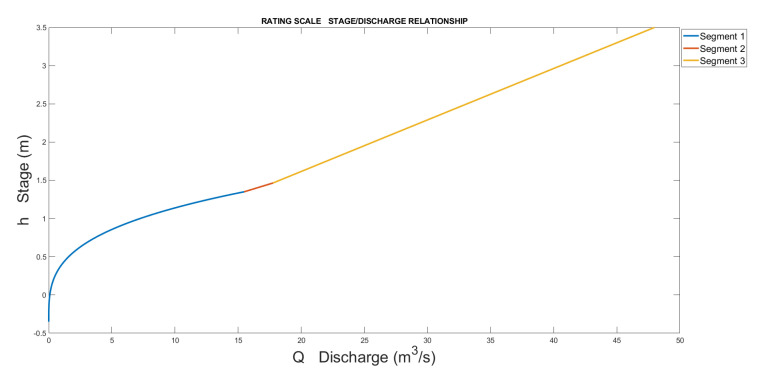
Rating scale of Nera with the 3 segments.

**Figure 17 sensors-23-02035-f017:**
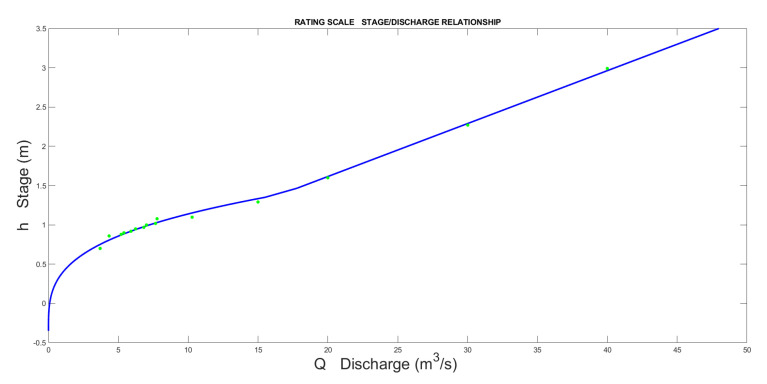
Complete rating scale of Nera with all the input points used for regression.

**Figure 18 sensors-23-02035-f018:**
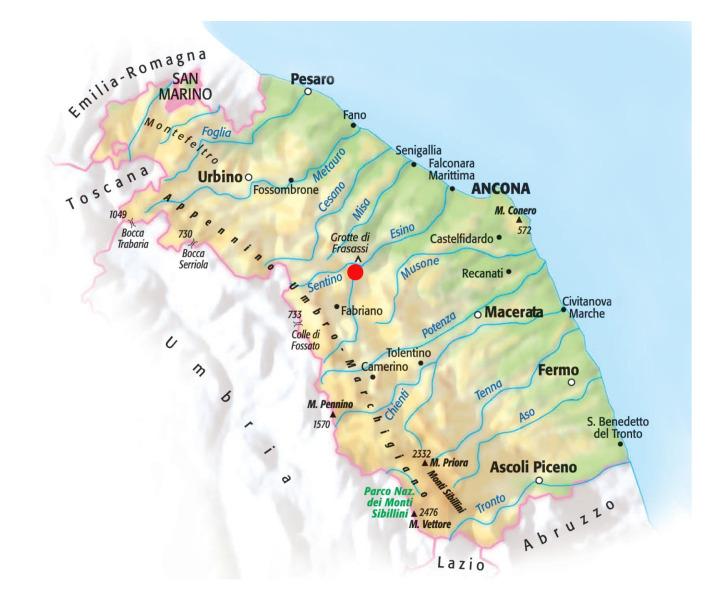
Marche Region, where the red dot indicates the investigated area of the Esino River located in the village of Camponocecchio, municipality of Genga (AN).

**Figure 19 sensors-23-02035-f019:**
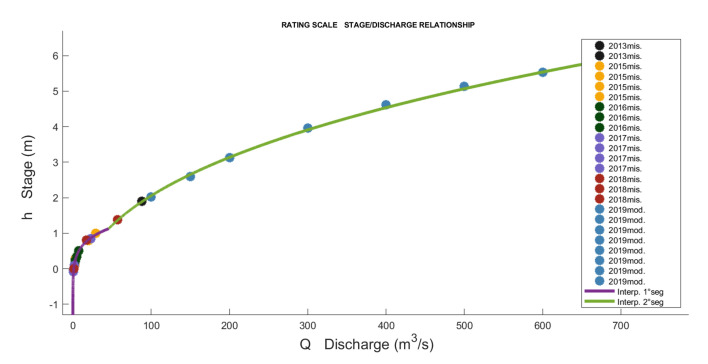
Rating scale of the Esino River with two segments.

**Table 1 sensors-23-02035-t001:** Sensors used for acquisitions in control stations.

Sensor Name	Sensor Type	Measurement Range	Frequency	Size
ETG ULS SENSOR	Ultrasonic	6 m, 12 m	65 kHz	5.5 kg, 380 × 150 × 150 mm
SITRANS Probe LU	Ultrasonic	6 m, 12 m	54 kHz	2.1 Kg, 198.9 × 130.1 × 139.7 mm
SITRANS Probe LR	Radar	0.3 m to 20 m	C-band, approx. 6 GHz	1.97 kg, 134 × 129 × 139 mm
SITRANS LR560	Radar	Up to 100 m	78 GHz FMCW	3.15 kg, 176 × 110 × 110 mm
VEGAPULS-SR-68	Radar	Up to 30 m	K-band (26 GHz technology)	17.2 kg, 420 × 120 × 92 mm

**Table 2 sensors-23-02035-t002:** Pontelatrave station on the Chienti River, section 2. HEC-RAS model: steady flow analysis, mixed flow regime, boundary conditions (upstream condition: slope of the water surface or bottom =0.0056363, downstream condition: slope of the water surface or bottom =0.0165812), and Manning’s roughness coefficient (riverbed =0.048, floodplains =0.05).

h (m)	Hydrometric Zero (m)	Stage *h* (m)	Discharge *Q* (m^3^/s)	Discharge *Q* (m^3^/s)	Error (%)
			(Input)	(Rating Scale)	
405.391	405.385	0.006	0.00001 (HEC-RAS model)	0.00001	0
405.615	405.385	0.23	0.806 (HEC-RAS model)	0.807	0.14
405.625	405.385	0.24	0.885 (HEC-RAS model)	0.884	−0.14
405.635	405.385	0.25	0.966 (measured)	0.964	−0.20
406.065	405.385	0.680	6.510 (measured)	6.510	0
406.250	405.385	0.865	10 (HEC-RAS model)	10.213	2.13
406.340	405.385	0.955	12 (HEC-RAS model)	12.094	0.78
406.540	405.385	1.155	17 (HEC-RAS model)	16.669	−1.95
406.640	405.385	1.255	20 (HEC-RAS model)	19.149	−4.25
407.130	405.385	1.745	35 (HEC-RAS model)	32.996	−5.73
407.680	405.385	2.295	50 (HEC-RAS model)	51.540	3.08
407.940	405.385	2.555	60 (HEC-RAS model)	61.306	2.18
408.180	405.385	2.795	70 (HEC-RAS model)	70.852	1.22

**Table 3 sensors-23-02035-t003:** Pontelatrave station models of the Chienti River.

Segment	Model	Model Validity Range	*R* ^2^	RMSE
Segment 1	$Q=18.129*{\left(h-0.005\right)}^{2.086}$	0.005≤h<0.252	1	0.002
Segment 2	$Q=13.644*{\left(h-0.005\right)}^{1.883}$	0.252≤h<0.841	1	0
Segment 3	$Q=14.775*{\left(h-0.075\right)}^{1.567}$	0.841≤h<3	0.99994	0.033

**Table 4 sensors-23-02035-t004:** Visso station on the Nera River. HEC-RAS model: steady flow analysis, mixed flow regime, boundary conditions (upstream condition: critical depth setting, downstream condition: critical depth setting), and Manning’s roughness coefficient (riverbed =0.03, floodplains =0.03).

Date	Stage *h* (m)	Discharge *Q* (m^3^/s)	Discharge *Q* (m^3^/s)	Error (%)	Discharge *Q* (m^3^/s)	Error (%)
		(Input)	(Rating Scale Used)		(Rating Scale Proposed)	
2 July 2011	0.7	3.71 (measured)	3.011	−18.84	3.158	−14.89
2 November 2017	0.86	4.35 (measured)	5.044	15.96	5.042	15.91
6 October 2017	0.88	5.21 (measured)	5.347	2.62	5.322	2.16
17 August 2017	0.9	5.39 (measured)	5.657	4.96	5.613	4.14
17 July 2017	0.92	5.9 (measured)	5.975	1.28	5.915	0.26
30 June 2017	0.95	6.23 (measured)	6.465	3.77	6.389	2.55
23 March 2018	0.97	6.81 (measured)	6.80	−0.15	6.719	−1.34
18 June 2018	1	7.02 (measured)	7.313	4.18	7.236	3.08
13 May 2017	1.02	7.67 (measured)	7.663	−0.09	7.596	−0.96
17 March 2017	1.08	7.77 (measured)	8.744	12.54	8.75	12.61
17 February 2017	1.1	10.28 (measured)	9.115	−11.34	9.161	−10.89
-	1.29	15 (HEC-RAS model)	12.854	−14.30	13.753	−8.32
-	1.6	20 (HEC-RAS model)	19.651	−1.75	19.767	−1.17
-	2.27	30 (HEC-RAS model)	36.475	21.58	29.704	−0.99
-	2.99	40 (HEC-RAS model)	56.941	42.35	40.400	1

**Table 5 sensors-23-02035-t005:** Visso station models of the Nera River.

Segment	Model	Model Validity Interval	*R* ^2^	RMSE
Segment 1	$Q=2.688*{\left(h+0.35\right)}^{3.3}$	−0.35≤h<1.35	0.9438	0.0931
Segment 2	$Q=19.650*{\left(h-0.56\right)}^{1.01}$	1.35≤h<1.465	0.9846	0.0585
Segment 3	$Q=14.732*{\left(h-0.26\right)}^{1.005}$	1.465≤h<3.5	0.9992	0.0153

**Table 6 sensors-23-02035-t006:** Section of Camponocecchio, Esino River. HEC-RAS model: steady flow analysis, mixed flow regime, boundary conditions (upstream condition: critical depth setting, downstream condition: critical depth setting), and Manning’s roughness coefficient (riverbed =0.05, floodplains =0.05).

Date	Stage *h* (m)	Discharge *Q* (m^3^/s)	Discharge *Q* (m^3^/s)	Error (%)	Discharge *Q* (m^3^/s)	Error (%)
		(Input)	(Rating Scale Used)		(Rating Scale Proposed)	
24 August 2017	−0.08	0.656	0.864	−31.71	0.803	−22.41
25 October 2017	−0.04	0.813	1.111	−36.65	0.961	−18.20
9 August 2018	0	1.23	1.377	−11.95	1.145	6.91
13 July 2015	0.1	2.26	2.125	5.97	1.742	22.92
26 June 2017	0.1	1.845	2.125	5.97	1.742	5.58
22 October 2016	0.22	3.18	2.125	−15.18	2.793	12.16
17 July 2013	0.28	2.871	3.16	0.63	3.496	3.50
15 December 2015	0.3	3.9	3.923	−0.59	3.762	-21.76
8 June 2016	0.34	4.75	4.492	5.43	4.345	8.53
30 May 2016	0.5	7.56	8.896	−17.67	7.512	0.63
2 February 2015	0.79	19.774	19.744	0.15	18.386	7.02
3 January 2018	0.8	17.5	20.159	−15.19	18.92	−8.11
3 February 2017	0.85	22.97	22.266	3.06	21.823	−8.11
28 February 2015	1	28.9	28.888	0.04	32.903	−13.85
24 February 2018	1.38	57	64.487	−13.14	57.962	−1.69
13 November 2013	1.9	88.1	114.543	−30.01	89.118	−1.16
-	2.02	100 (HEC-RAS model)	126.140	−26.14	97.53	2.47
-	2.60	150 (HEC-RAS model)	182.348	−21.57	145.16	3.23
-	3.12	200 (HEC-RAS model)	230.760	−15.38	198.401	0.80
-	3.96	300 (HEC-RAS model)	393.176	−31.06	307.618	−2.54
-	4.62	400 (HEC-RAS model)	577.626	−44.41	415.421	−3.86
-	5.13	500 (HEC-RAS model)	742.824	−48.56	513.058	−2.61
-	5.53	600 (HEC-RAS model)	883.876	−47.31	598.882	0.19
-	5.82	700 (HEC-RAS model)	991.794	−41.68	666.882	4.73

**Table 7 sensors-23-02035-t007:** Camponocecchio station models of the Esino River.

Segment	Model	Model Validity Interval	*R* ^2^	RMSE
Segment 1	$Q=0.047*{\left(h+1.59\right)}^{6.88}$	−1.59≤h<1.27	0.988	0.14
Segment 2	$Q=3.069*{\left(h+1.61\right)}^{2.683}$	1.27≤h<6	0.998	0.03

## Data Availability

The data presented in this study are available from the corresponding author upon reasonable request.

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
