# Peer review of "Discharge Monitoring in Open-Channels: An Operational Rating Curve Management Tool"

_sensors, 2023, doi:10.3390/s23042035_

Round 1

Reviewer 1 Report

Abstract

I suggest expanding this part a bit, by providing a more complete overview of the study, and what are the key messages deriving from your work. This is needed to engage readers from the very beginning.

Introduction

The description of the local issue is pretty clear and comprehensive, but a national/international picture is lacking. Why is this study important besides the very local situation of the Marche Region? I suggest adding a few words about other systems applied in Italy/Europe.

Materials and Methods

Is Fig.1 relevant? If so, please acknowledge the source/authors, and the location.

Stage-discharge relationships have extensive literature. I suggest expanding the discussion of what is new in your approach, and what are you taking from other approaches/datasets. In the present version, it is not straightforward catching what are you actually developing, and what is taken from existing material.

The model equation

I suggest rephrasing the title of this section to something like "Modelling approach".

Fig. 3: as for fig. 1, I suggest adding the location.

I suggest combining Figs 4 and 5 in the same figure with two panels, as this can allow readers for a more immediate comparison.

Data acquisition

Fig. 6: what about adding the dimension of the sensor?

Please provide more references to all the equations, if their definition derives from the literature. This can help readers in having a better picture of the assumptions behind them.

Results

I am not sure that a single case study can be enough to prove the reliability of the approach. If feasible, I suggest looking at another river, possibly with different characteristics. This can help very much in proving the quality of your approach, and also its applicability to different contexts.

In addition, a better description of the study area, with a map, can help readers not familiar with the Marche Region.

Discussion

Table 2: Is Pontelatrave a river or a city or a station? Please be consistent. 

I was not able to find any information about the HEC-RAS model that you used as a comparison. I think that more details on this model are needed, as some sources of uncertainties can also derive from the modelling assumptions.

This section should be expanded also discussing the transferability of your approach. Indeed, as already stated, the focus of your work is very local, and it's hard to figure out the usefulness of your approach in rivers with other characteristics.

Conclusions

Are figs. 13-15 and tables 4-5 part of this section? If not, I suggest keeping them out to increase the readability.

Reviewer 2 Report

The manuscript is well detailed. I am attaching my comments to improve it further.

Round 2

Reviewer 1 Report

Dear Authors,

I would like to thank you for having answered all my doubts.

In my opinion, the present version can be processed further.